# Evaluating the User Experience and Usability of the MINI Robot for Elderly Adults with Mild Dementia and Mild Cognitive Impairment: Insights and Recommendations

**DOI:** 10.3390/s24227180

**Published:** 2024-11-08

**Authors:** Aysan Mahmoudi Asl, Jose Miguel Toribio-Guzmán, Álvaro Castro-González, María Malfaz, Miguel A. Salichs, Manuel Franco Martín

**Affiliations:** 1Psycho-Sciences Research Group of IBSAL, University of Salamanca, 37007 Salamanca, Spain; jmtg@usal.es (J.M.T.-G.); mfrancom@saludcastillayleon.es (M.F.M.); 2Department of Research and Development, Iberian Institute of Research in Psycho-Sciences, INTRAS Foundation, 49024 Zamora, Spain; 3Personality, Evaluation and Psychological Treatments Department, University of Salamanca, 37007 Salamanca, Spain; 4Robotics Lab, Department of System Engineering and Automation, Universidad Carlos III de Madrid, 28911 Madrid, Spain; acgonzal@ing.uc3m.es (Á.C.-G.); mmalfaz@ing.uc3m.es (M.M.); salichs@ing.uc3m.es (M.A.S.); 5Psychiatry and Mental Health Service, Assistance Complex of Zamora, 49021 Zamora, Spain

**Keywords:** usability, user experience, social robots, dementia, mild cognitive impairments

## Abstract

**Introduction**: In recent years, the integration of robotic systems into various aspects of daily life has become increasingly common. As these technologies continue to advance, ensuring user-friendly interfaces and seamless interactions becomes more essential. For social robots to genuinely provide lasting value to humans, a favourable user experience (UX) emerges as an essential prerequisite. This article aimed to evaluate the usability of the MINI robot, highlighting its strengths and areas for improvement based on user feedback and performance. **Materials and Methods**: In a controlled lab setting, a mixed-method qualitative study was conducted with ten individuals aged 65 and above diagnosed with mild dementia (MD) and mild cognitive impairment (MCI). Participants engaged in individual MINI robot interaction sessions, completing cognitive tasks as per written instructions. Video and audio recordings documented interactions, while post-session System Usability Scale (SUS) questionnaires quantified usability perception. Ethical guidelines were followed, ensuring informed consent, and the data underwent qualitative and quantitative analyses, contributing insights into the MINI robot’s usability for this demographic. **Results**: The study addresses the ongoing challenges that tasks present, especially for MD individuals, emphasizing the importance of user support. Most tasks require both verbal and physical interactions, indicating that MD individuals face challenges when switching response methods within subtasks. These complexities originate from the selection and use of response methods, including difficulties with voice recognition, tablet touch, and tactile sensors. These challenges persist across tasks, with individuals with MD struggling to comprehend task instructions and provide correct answers and individuals with MCI struggling to use response devices, often due to the limitations of the robot’s speech recognition. Technical shortcomings have been identified. The results of the SUS indicate positive perceptions, although there are lower ratings for instructor assistance and pre-use learning. The average SUS score of 68.3 places device usability in the “good” category. **Conclusions**: Our study examines the usability of the MINI robot, revealing strengths in quick learning, simple system and operation, and integration of features, while also highlighting areas for improvement. Careful design and modifications are essential for meaningful engagement with people with dementia. The robot could better benefit people with MD and MCI if clear, detailed instructions and instructor assistance were available.

## 1. Introduction

Social robots are commonly known as autonomous robots that possess human-like characteristics and engage in social interactions with humans [1]. Over the past few years, there has been a rising need for social robots in sectors such as healthcare, service [2], education [3], and entertainment. This surge in demand has driven a significant advancement in this technology [4]. We are currently witnessing an increase in the number of new social robots, accompanied by research initiatives from both academic and industry areas.

Social robots have found diverse applications, catering to a broad spectrum of human demographics spanning from children to the elderly. This reveals that social robots have been favourably received and embraced by the elderly adults, unlike in other areas such as children’s education [5].

In recent times, there has been a notable focus on the utilization of social robots for PwD and MCI [6,7]. Social robots offer a ray of hope for PwD by providing companionship and engagement, alleviating feelings of isolation and loneliness [7]. These robots can enhance cognitive functions through interactive games, memory prompts, and personalized activities, contributing to the overall well-being of PwD [8]. With their ability to provide reminders for medication, appointments, and daily routines, social robots empower PwD and MCI to maintain a sense of independence and control over their lives. Social robots can reduce caregiver burden by assisting in routine tasks, allowing caregivers to focus on providing emotional support and strengthening their relationship with the person with dementia [9]. Through stimulating interactions, reminiscence therapy, and emotional support, social robots create a conducive environment for improved emotional states and enhanced communication.

The User Experience (UX) and usability of these robots are rooted in their intuitive interfaces, enabling users to communicate and interact seamlessly. This blurs the boundaries between humans and machines, ensuring overall satisfaction and the system’s effectiveness. Through advanced sensors and artificial intelligence, social robots can perceive and respond to human gestures, expressions, and speech, creating an immersive and personalized UX. This tailored interaction enhances engagement, making tasks like companionship, education, and assistance more enjoyable and efficient [10]. The evolving design of these robots focuses on user-centred principles, aiming to minimize complexity and optimize ease of use. As a result, the realm of social robotics continues to redefine the boundaries of technological interaction by creating intuitive, meaningful, and emotionally resonant experiences for users across diverse contexts.

Even though interacting with social robots can be entertaining, it could pose specific challenges for PwD and MCI, stemming from problems such as memory and language deficits and their special needs. Careful consideration of design and interaction methods, as well as ongoing customization, is essential to address challenges and ensure that social robots effectively support and engage this target group in a meaningful and respectful manner [11]. Assessing the usability and UX of the robot agent should commence early in the development process, enabling adjustments or substitutions in its attributes and operations to ensure a favourable UX prior to market launch [12].

Lately, a robotic system named MINI has been designed and developed within the Robotics Lab research group at Carlos III University in Madrid. Its primary objective is to assist seniors, offering cognitive and social engagement to elderly individuals affected by neurodegenerative disorders, particularly those dealing with cognitive decline [13]. The robot has the ability to initiate user engagement using both spoken and non-spoken methods. This compact, desktop robot is enveloped in synthetic fur and furnished with a speech recognition mechanism, with touch-sensitive areas on its shoulders and heart, an RGB-D camera for assessing depth in the surroundings, expressive uOLED eyes, and a tablet-based interface serving as an input and output mechanism to showcase software (Version 2) content. The robot’s head and arms have movement restricted to a single dimension [14,15] (see Figure 1).

A series of games and activities are designed and integrated into the robotic platform. Figure 2 demonstrates an overview of the app and games provided by the MINI robot. The service has been designed in a co-creation process and is centred on the needs and preferences of elderly adults and to fulfil the goal of a social companion for the MINI robot.

## 2. Materials and Methods

### 2.1. Participants and Settings

This mixed-methods study employed a total of ten people, 5 with MD and 5 with MCI (M = 77.3 years and SD = 4.9). The suggested inclusion criteria were as follows: age 65 and above, being able to make decisions, being able to read and write, and not having any sense loss that would prevent the usage of the devices (e.g., blindness or deafness). Patients from the INTRAS Foundation’s Memory Clinic and memory workshops (Fundacíon Intras, Zamora Spain) were recruited to participate in the study, and the trial was undertaken within the controlled environment of the INTRAS Foundation’s usability laboratory. Ethical approval for this research was previously obtained from Zamora Health Area Medication Research Ethics Committee, under registration number 574.

### 2.2. Procedure of the Field Test

Participants and/or their next of kin provided informed consent and socio-demographic information via a questionnaire before the test. Following that, the evaluator explained the approach and tasks associated with the MINI robot interaction. The test methodology required the engagement of just one participant at a time, allowing for rigorous monitoring of their task performance and behaviour. The evaluation consisted of a series of predetermined activities divided into subtasks. Following the completion of these tasks, participants were invited to complete a questionnaire designed to assess the usability of the MINI robot. Notably, the field tests were thoroughly recorded by audio and video recordings, allowing for further thorough examination. The following protocol was adhered to throughout the users’ interaction with the robot:
The user initiated the test by sitting before the robot and interacting after receiving task instructions. The test was initiated with the robot in a dormant state.Interaction with MINI began with the user waking the robot with the phrase: “Hello MINI!” Upon activation, the robot introduced itself to familiarize the user and allay any initial fears.Subsequently, the robot inquired about the user’s intended actions, with participants adhering closely to the provided instructions (conveyed on A4-sized sheets). The session encompassed nine distinct tasks, each employing varied forms of interaction, including voice interaction (VI) and touch interaction (TI).
The chosen tasks and interaction methods are as follows:
**Task 1.** *Wake up the robot*. The participant has to wake up the robot by saying, “Hello, MINI” (VI).
**Task 2.** *See photos of my city*. See photos (VI) > From my city (VI). The robot shows, through the tablet, a series of pictures of the city. Once the task is over, the robot asks, “Did you like the task?” (TI).
**Task 3.** *Guessing Meals*. Entertainment (TI) > Games (TI) > Guessing Meals (VI). Pictures are shown, and different regional dishes are asked about.
**Task 4.** *Stop the task*. Before the previous task ended, the participant had to stop the task, by tapping the robot on the shoulder. Then, the robot would ask, “Do you want to continue with the task?” (TI) or “Why did you stop the task?” (TI).
**Task 5.** *Weather news.* News (VI) > Weather news (TI). The robot gives the weather forecast for the area. Once the task is over, the robot asks, “Did you like the task?” (TI).
**Task 6.** *Robot suggestion.* The robot suggests a task to the participant (IT or IV). Once the task is over, the robot asks, “Did you like the task?” (TI).
**Task 7.** *Relaxing music*. Music > Relaxing music. The robot plays a piece of relaxing music.
**Task 8.** *Stop the task*. Before the previous task ended, the participant had to stop the task again (TI) by tapping the robot on the shoulder. Then, the robot would ask, “Do you want to continue with the task?” (TI) or “Why did you stop the task?” (TI).
**Task 9.** *Go to sleep*. when the robot asks. “What do you want to do?” the participant must say, “Go to sleep” (VI).

### 2.3. Method of Data Collection

Data collection consisted of three phases: the administration of pre-test questionnaires to gather socio-demographic information, direct observations and video recordings, and a post-test evaluation through questionnaires.

Sociodemographic survey: this was used to gather relevant sociodemographic details from participants, including factors such as age, gender, educational background, extent of cognitive impairment, and utilization of information and communication technology (ICT).

The System Usability Scale (SUS): a widely employed questionnaire-based tool developed by John Brooke in 1986 to evaluate the perceived usability of diverse products, services, and systems, spanning softwares, websites, applications, and technological devices. This approach has gained significant popularity in assessing user satisfaction and experience. The SUS (Appendix A) utilizes a straightforward 10-item Likert scale questionnaire, systematically designed to capture users’ subjective views on usability dimensions like ease of use, efficiency, learnability, and user-friendliness. Respondents rate their agreement level with statements related to the evaluated system. Post-questionnaire scores are aggregated to form a usability score between 0 and 100, with higher scores indicating better perceived usability. The SUS serves as a quantitative tool to compare user satisfaction and usability across various systems, aiding in informed decision-making for enhancing the UX. In essence, the System Usability Scale is a vital resource for evaluating user perceptions and guiding design and usability improvements for more user-friendly products and systems [16]. The SUS test yields a score between 0 and 100, indicating how useful a device is for the user. A score of 68 points or above indicates good usability. According to a recent scoping review [17], the SUS is the most commonly used questionnaire for evaluating the usability of the social robots.

### 2.4. Data Analysis

During the field test, observation notes were recorded. Both direct and video observations were conducted to collect data on user performance, time to complete tasks, success rates, failure rates, incidents, user and robot behaviours, and other pertinent information. The participants completed the SUS at the end of the interaction session and the preliminary scores for the SUS were provided by the main researcher, who analyzed the data. The scores for SUS were presented using basic descriptive statistics for each item, including mean scores, standard deviation, range, and mode. Furthermore, a total SUS score was computed for each participant, and the mean of all SUS scores was derived.

## 3. Results

### 3.1. User Testing; Observations

An assessment of several elements influencing correct task performance was considered as follows (Table 1):-**Difficulty in Understanding the Exercise Methodology (DEM):** User misunderstands the exercise instructions.-**Difficulty in Recognizing the Response Method (DRM):** User is unsure which device to use for responses (e.g., the user tries to answer verbally when the response method is through the buttons on the tablet screen.).-**Difficulty in Using the Device Corresponding to the Response Method (DUR):** User struggles with using the response device or the robot fails to understand the user.-**Difficulty with the Content of the Response (DCR):** User does not know the correct answer or how to respond.

These difficulties may arise at the beginning of each task and, depending on the case, persist during the test. Therefore, it is important to take into consideration the help that users may have received in the different tasks.

Table 2 indicates the percentage of participants that needed assistance from the instructor at the start of each task with “comprehending the written instructions of each task” and “response method (verbal or touch)”. As can be observed, the percentages of required help from the instructor in people with MD are often higher than in MCI. Furthermore, both people with MD and MCI needed more help with the response method than with the instructions. Tasks that required reacting via the robot’s touch sensors (4a and 8a) needed more help in both groups.

The majority of the tasks required both verbal and physical interactions. Table 2 shows that when there is a shift in response method between subtasks, individuals with MD have greater issues and require more assistance.

Regarding the response method, as explained in the difficulties that may arise, it is necessary to distinguish between problems caused by the user not knowing exactly which devices he or she has to answer on (DRM) (for example, waiting for the buttons to appear on the screen when in fact he or she has to answer by voice) and problems caused by the use of the response device itself (DUR). Considering the second aspect, the following issues were noticed:Microphones (voice): Because the MINI’s speech recogniser is not always turned on, users must wait until the robot beeps before responding. This caused confusion since many users responded too quickly. There were also instances where users did not vocalize correctly (e.g., too soft voice, spelling the answer) resulting in the robot being unable to comprehend them, even though they reacted adequately to what was asked of them.Screen–Tablet (touch): Although this device did not cause any problems, there were some instances where users did not tap the screen buttons properly because they did not tap it with their fingertips, but with their fingernails, or they pressed it too quickly and loosely, causing the tablet to not process it.Touch sensors: The main issue with this kind of response was that users had difficulty determining where they needed to touch the robot in order to activate the sensors.

Evidently, all these issues manifested themselves equally during task completion. The level of success achieved by the distinct groups in each task, along with the corresponding challenges that emerged, are detailed in Table 1. On one hand, it is evident that the MD group encountered challenges mainly linked to response uncertainty (DCR) and complexities in grasping the task methodology (DEM), rather than the actual response method (DRM). The MCI group encountered similar difficulties to the MD group, albeit with more pronounced struggles in using the response device (DUR). Importantly, a noteworthy portion of these challenges stemmed from the robot’s struggles in comprehending the provided answer. Furthermore, specific technical deficiencies were identified as a result of our analysis (Table 3). It is essential to clarify that we are only disclosing the game-specific issues detected during the field test. It is possible that additional imperfections existed within other games that were not selected and played by the user during this particular testing phase.

### 3.2. System Usability Scale

To provide a more comprehensive examination of the SUS score, an item-by-item analysis was conducted to determine the direction of participants’ responses. The descriptive data for the items can be found in Appendix A. The items were both positively and negatively oriented. Negative items were numerically inverted so that the mean, on a scale of 1–5, represents the lowest to highest level of agreement for all items.

Figure 3 shows the SUS questionnaire mean scores for each item. It can be seen that the majority of the ratings supported the usability of the MINI robot. Of particular note are the items that received the highest levels of agreement, including the users’ belief that the majority of people would quickly learn how to use it (4.40), that the system was not cumbersome to use (4.40), that they will use it frequently (3.90), that the system’s various features were well integrated (3.90), and that there was no perception of excessive complexity in the system (3.90).

The lowest ratings were assigned to the need for instructor assistance in operating the robot (3.00) and the need to learn many things before utilizing the system (3.20). There were no negative ratings in the mean scores (3>).

The SUS test gives a total score between 0 and 100, indicating how useful a device is to the user. A score of 68≤ indicates favourable usability. With an average SUS score of 68.3 out of 100, the device falls into the “good” usability category (68–80.3).

## 4. Discussion

In this study, we examined the UX and usability of the MINI robot and the challenges that might arise during human–robot interactions with individuals with MD and MCI in laboratory settings. The qualitative and quantitative data gathered from the field test observations and video analysis revealed several facets of the robot interface that should be considered when employed to serve these target groups.

Based on our observational data, we identified several challenges users face when interacting with social robots. These include difficulties in understanding exercise instructions (DEM), recognizing the appropriate response method (DRM), using the correct device for responses (DUR), and knowing the content of the response (DCR). Therefore, they relied on the instructor assistance for each task, for comprehending written instructions and responding appropriately, whether verbal or tactile.

Tasks requiring interaction via the robot’s touch sensors (tasks 4a and 8a) showed an increased need for assistance in both groups, underscoring the inherent complexity of touch-based interactions. Users’ difficulties in completing these tasks could be attributed to unfamiliarity with the robot, uncertainty about how and where to touch, and confusion when shifting between verbal and touch interaction methods. This highlights the heightened challenge posed by having two interaction channels (touch and speech) and the overall complexity of touch-based interactions. Similar studies have documented these challenges in human–robot interactions. For instance, research by Leite et al. (2013) [18] found that users often struggle with touch-based tasks due to the lack of clear tactile feedback and the ambiguity of touch-sensitive areas on the robot. Additionally, the study by Dragan and Srinivasa (2013) [19] highlighted that users frequently experience difficulty when required to switch interaction modalities, such as from speech to touch. This transition can be cognitively demanding, particularly for individuals with motor or cognitive impairments. Addressing these challenges requires a multifaceted approach, including better designing touch interfaces, incorporating clearer instructions for users, and adaptive support systems that can help users navigate multiple interaction modalities seamlessly.

Turning attention to the response method itself, as elaborated upon in the context of potential difficulties, it is imperative to differentiate between challenges arising from user confusion, recognizing response methods (DRM), and challenges stemming from the use of the response device (DUR). Regarding the issues encountered in the response method, we noticed that the majority of them were caused by the users’ confusion using the response device, or the robot having difficulty recognizing the answer due to its limitations in comprehending speech. Issues regarding the microphones, touch sensors, and tablet interface emerged in similar studies with social robots [8,20]. The poor speech recognition system hasbeen listed as the main usability issue in most research on social robots with speech capabilities [7,8,20,21]. So far, social robots lack the ability to engage in seamless conversations with users. At the same time, our target population is not used to such conversational agents. Consequently, there is an urgent need for significant improvements in the speech recognition system. It is therefore imperative to incorporate an initialisation or training phase to ensure optimal and efficient use of the robot.

Concerning task performance, particularly in the MD group, difficulties are more often caused by not knowing the correct answer (DCR) and difficulty understanding the task methodology (DEM) than by the subtleties of the response method (DRM). Hence, our suggestion is to provide explicit usage instructions using uncomplicated language, ensuring readability and accessibility for individuals with MD and MCI. Another study of the MINI robot [22] found that stakeholders preferred activities that were not similar to quizzes to avoid potential conflicts with cognitive challenges. However, some participants in this study expressed their readiness for moderately challenging tasks. Consequently, it is essential for developers to adjust the difficulty levels of task content to accommodate the cognitive status and preferences of users. This was previously suggested by the developers of the MINI robot [23,24] regarding personalisation of human–robot interaction.

The challenges faced by the MD and MCI groups highlight the intricate relationship between cognitive abilities, device operability, and the overall success of human–robot interactions. These findings emphasize the crucial importance of designing user-friendly interfaces, clear user instructions, and interpretive capabilities within robotic systems to improve overall usability and facilitate seamless user engagement.

The overall SUS score indicates a good usability level, verifying the system’s effectiveness and alignment with user expectations. The majority of participants perceived minimal difficulty with learning to use the robot, operational complexity, and interaction with the MINI robot, and would like to use the robot frequently. Additionally, they acknowledged the smooth integration of system functionalities and perceived the system’s complexity as manageable. Less favourable ratings were assigned to the perceived need for instructor assistance in operating the robot and the need for pre-use learning. This aligns with their performance and accomplishments and the obstacles they encountered in achieving tasks during the field test. Relatively lower scores in feeling confident operating the robot, the robustness of the robotic system, and perceived ease of use may be mainly due to technical issues and design problems inherent in the robot. In another study [25] investigating the acceptance of the MINI robot, the perceived ease of use ratings after one month of interaction were similar to or slightly higher than those in our current study. This observation suggests that prolonged interaction may potentially increase perceived ease of use.

Based on our results and observations, we propose a list of modifications for future versions of the robot (Table 4). As MINI is a research platform under continuous development, these suggestions aim to enhance the user experience when interacting with the robot. Given that MINI’s verbal communication relies on available commercial softwares and solutions for speech recognition and production [13], our recommendations for the speech interface can be applied to other smart devices, such as smart speakers and smartphones, to improve their verbal interaction capabilities.

## 5. Limitations of the Study

Despite these promising findings, several limitations in this study warrant attention. First, our study involved a sample of 10 participants. While this number of participants was sufficient to conduct a usability test, from a scientific perspective [26], its potential impact on the broader generalisability of our findings warrants consideration. Secondly, this study was conducted using specific apps and games that were predefined. Therefore, the results should be interpreted with caution when projecting to different versions of the robot with alternative apps and games. Different features of other apps and games could potentially influence the usability dynamics of the robot.

## 6. Conclusions

In conclusion, our study deeply examines the MINI robot’s UX and usability challenges. The robot has strengths in easy learning, simple operation, and seamless feature integration, but also highlights areas for improvement, including the general system, games and apps, and speech and voice related features. The authors assume that using the MINI robot effectively requires clear and detailed instructions and instructor assistance. By making suggested changes and listening to users, developers can improve the robot’s usability and UX, ensuring the overall effectiveness. To overcome challenges, careful design and interaction methods, along with ongoing adjustments, are essential. This process can turn the MINI robot into a more user-centred system.

## Figures and Tables

**Figure 1 sensors-24-07180-f001:**
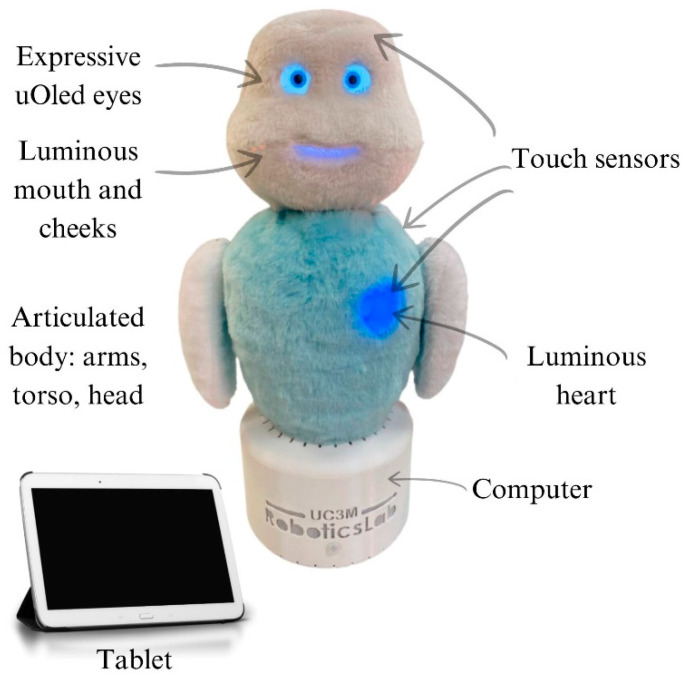
MINI robot components.

**Figure 2 sensors-24-07180-f002:**
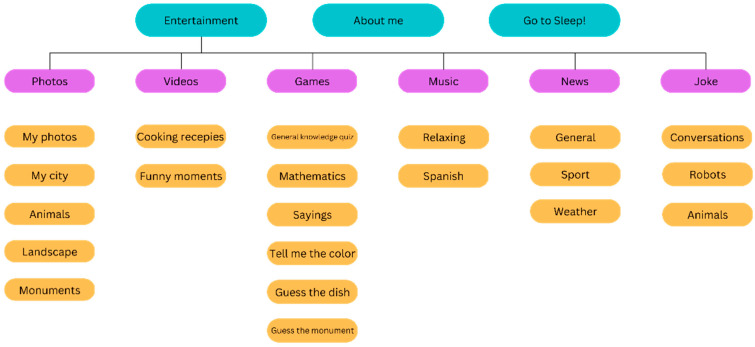
MINI robot apps categorized: blue for general categories, purple for entertainment types, and orange for specific activities within each.

**Figure 3 sensors-24-07180-f003:**
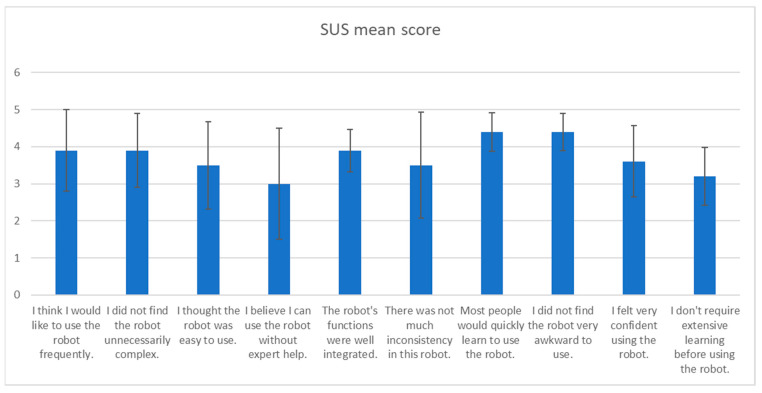
SUS score for each Item.

**Table 1 sensors-24-07180-t001:** Percentage of success and difficulties encountered for each task.

			MCI					MD				
Task		R	%SC	%CD	Dif.	%PC	Dif.	%SC	%CD	Dif.	%PC	Dif.
1	Wake up MINI	V	60.0	40.0	40.0% DEM	0.0	-	100.0	0.0	-	0.0	-
2	Look at photos	V	40.0	40.0	40.0% DUR	20.0	20.0% DUR	60.0	40.0	20.0% DUR 20.0% DRM	0.0	-
	Photos of my city	V	60.0	0.0	-	40.0	40.0% DUR	80.0	0.0	-	20.0	20.0% DUR
	Did you like the task?	T	60.0	40.0	40.0% DUR	0.0	-	0.0	60.0	20.0% DRM 40.0% DUR	40.0	40.0% DUR
3	Entertainment	T	40.0	60.0	40.0% DRM 20.0% DUR	0.0	-	80.0	20.0	20.0% DRM	0.0	-
	Games	T	60.0	20.0	20.0% DEM	20.0	20.0% DEM	60.0	40.0	20.0% DEM 20.0% DRM	0.0	-
	Guess food	V	20.0	20.0	20.0% DRM	60.0	40.0% DRM 20.0% DEM	40.0	40.0	20.0% DRM	20.0	20.0% DCR
4	Pause the task	T	20.0	80.0	80.0% DEM	0.0	-	0.0	100.0	100.0% DEM	0.0	-
	Do you want to continue with the task?	T	100.0	0.0	-	0.0	-	80.0	20.0	20.0% DRM	0.0	
	Why did you stop the task?	T	40.0	20.0	20.0% DCR	40.0	40.0% DCR	20.0	0.0	-	80.0	80.0% DCR
5	News	V	60.0	40.0	20.0% DUR 20.0% DRM	0.0	-	100.0	0.0	-	0.0	-
	Weather news	T	60.0	40.0	40.0% DUR	0.0	-	60.0	0.0	-	40.0	40.0% DEM
	Did you like the task?	T	40.0	60.0	20.0% DUR 40.0% DCR	0.0	-	80.0	20.0	20.0% DRM	0.0	-
6	Robot’s randomly suggested task	V/T	60.0	20.0	20.0% DCR	20.0	20.0% DEM	60.0	40.0	40.0% DCR	0.0	-
	Did you like the task?	T	20.0	20.0	20.0% DUR	60.0	60.0% DCR	60.0	0.0	-	40.0	40.0% DCR
7	Music	V	60.0	40.0	40.0% DUR	0.0	-	80.0	0.0	-	20.0	20.0% DEM
	Relaxing music	T	60.0	0.0	-	40.0	40.0% DEM	100.0	0.0	-	0.0	-
8	Pause the task	T	40.0	60.0	60.0% DEM	0.0	-	20.0	80.0	80.0% DEM	0.0	
	Do you want to continue with the task?	T	100.0	0.0	-	0.0	-	80.0	20.0	20.0% DCR	0.0	
	Why did you stop the task?	T	60.0	20.0	20.0% DCR	20.0	20.0% DCR	40.0	0.0	-	60.0	60.0% DCR
9	Go to sleep	V	20.0	40.0	40.0% DUR	40.0	40.0% DRM	80.0	0.0	-	20.0	NA

SC: successfully completed; CD: completed with difficulty; PC: partially completed; DCR: difficulty with the content of the response; DEM: difficulty in understanding the methodology of the exercise; DRM: difficulty in recognizing the response method; DUR: difficulty in using the response method; R: Response; T: Touch; V: Verbal; Dif.: Difficulty.

**Table 2 sensors-24-07180-t002:** Percentage of participants who required help from the instructor in the completion of each task.

			MCI (%)		MD (%)		Total (%)	
			Help from Instructor		Help from Instructor		Help from Instructor	
Task	Subtask	Form of Required Response	Comprehending the Instructions	Response Method	Comprehending the Instructions	Response Method	Comprehending the Instructions	Response Method
1	Wake up MINI	V	0.0	20.0	0.0	20.0	0.0	20.0
2	a. Look at photos	V	0.0	40.0	40.0	60.0	20.0	50.0
	b. Photos of my city	V	0.0	0.0	0.0	20.0	0.0	10.0
	c. Did you like the task?	T	20.0	20.0	60.0	40.0	40.0	30.0
3	a. Entertainment	T	40.0	40.0	40.0	60.0	40.0	50.0
	b. Games	T	20.0	20.0	0.00	60.0	10.0	40.0
	c. Guess food	V	20.0	20.0	40.0	40.0	30.0	30.0
4	a. Pause the task	T	60.0	60.0	100.0	100.0	80.0	80.0
	b. Do you want to continue with the task?	T	20.0	80.0	20.0	60.0	20.0	70.0
	c. Why did you stop the task?	T	0.0	40.0	0.0	80.0	0.0	60.0
5	a. News	V	0.0	20.0	20.0	40.0	10.0	30.0
	b. Weather news	T	20.0	0.0	20.0	0.0	20.0	0.0
	c. Did you like the task?	T	0.0	0.0	0.0	20.0	0.0	10.0
6	a. Robot’s randomly suggested task	V/T	0.0	0.0	0.0	40.0	0.0	20.0
	b. Did you like the task?	T	0.0	60.0	0.0	60.0	0.0	60.0
7	a. Music	V	20.0	40.0	40.0	60.0	30.0	50.0
	b. Relaxing music	T	0.0	0.0	0.0	60.0	0.0	30.0
8	a. Pause the task	T	20.0	40.0	60.0	60.0	40.0	50.0
	b. Do you want to continue with the task?	T	20.0	20.0	20.0	20.0	20.0	20.0
	c. Why did you stop the task?	T	0.0	0.0	0.0	40.0	0.0	20.0
9	a. Go to sleep	V	0.0	40.0	0.0	60.0	0.0	50.0

T: touch; V: verbal.

**Table 3 sensors-24-07180-t003:** MINI robot related issues.

General Issues	Game Specific Issues
The Robot …	The Robot …
… was unable to recognize speech. … froze. … does not close its eyes in sleeping mode. … wakes up and starts another session without being called or touched. … says “I did not hear you well” in the case that the user is quiet and not answering verbally at all. … does not recognize the response of the user in the case that they say “I do not know” and keeps repeating the question.	… does not give enough time to calculate the math operation. … does not speak fluently asking mathematical questions (instructor had to repeat the question). … does not give enough time to order the words in the “Sayings” game. … during the “Tell me the colour” game, asks for a matching colour and shows the photos on the tablet with delay. … in the “guess monument” and “guess the food” games, asks questions and does not provide the answers options immediately.

**Table 4 sensors-24-07180-t004:** Suggested modifications to the MINI robot.

Suggestion
- Upgrade speech recognition: Enhance the system’s ability to accurately capture and interpret user speech. - Prompt listening: Allow the robot to immediately listen and identify user speech after asking a question. - Recognize uncertainty: Enable the robot to recognize phrases like “I don’t know” and provide an appropriate response. - Improve vocal output: Enhance the smoothness and clarity of the robot’s speech.
- Extend response time: Increase the time limit for responding in games like “Mathematics” and “Sayings,” which require more time to guess the answer. - Introduce difficulty levels: Design different difficulty levels for applicable games, such as the “General Knowledge Quiz.”
- Synchronize speech and display: Ensure seamless alignment between the robot’s spoken content and the information displayed on the tablet interface. - Remove unrelated behaviours: Eliminate any undesired or unrelated behaviours exhibited by the robot.

## Data Availability

Some data are not publicly available because the data contains information that could compromise research participant privacy.

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
