# Peer review of "Evaluating the User Experience and Usability of the MINI Robot for Elderly Adults with Mild Dementia and Mild Cognitive Impairment: Insights and Recommendations"

_sensors, 2024, doi:10.3390/s24227180_

Round 1
Reviewer 1 Report
Comments and Suggestions for Authors
Thank you for your paper.
1. Please explain in the paper why each user only utilized the system once, given that you pointed out that reuse could change the perceived ease of use issues. (reference to use one month later)
2. In the description of tasks, there are references to "stop the test" ... please describe in the paper if the "tasks" were explained as "tests" or "tasks" and whether the change in the terms led to confusion for the subjects. (this shows up in more than one place)
3. Please change the order of discussion of Table 2 and Table 1 or the order of the tables, so that the first discussion talks about the first table the user will find. (ie whatever the first table to be discussed should be called Table 1, etc)
4. In the tables please put the legend for the table with the title of the table (which is at the top) rather than at the bottom of the table itself.
5. Please clarify in the paper, whether "...the study employed ten people with MD and MCI..." (ln 111), involves 5 and 5 or 10 and 10.
6. Please change micros to microphones (ln 229 and 314)
7. Please check if Trial Phase (ln321) is referring to a training phase; if it is trial phase, please define in the paper what is intended.
8. Please change the acronyms to be more consist with the words before them...eg "...not know the correct answer (DCR).." [use Correct Response or DCA?] etc; subtleties of the response method (DRM)... there are a number of these and aligning the words with the letters would be helpful. (or change the words to be consistent with the acronyms)
9. please clarify in the paper who provided the "preliminary scores for SUS"and how this was done. (ln 189)
10. In Figure 3, please provide bars to indicate the variability about the means
11. Please explain in the paper the basis for saying "...MINI's verbal communications rely on state of the art software..." (ln 356)
12. Please supply the supplementary data referred to in ln 262. (the only supplementary data I found was the DEC 1986 questionaire; I'm assuming you are not referring to that.)
13. Please include in the paper support for the conclusion: "This comprehensive approach ensures social robots can effect effectively engage individuals with dementia in a meaningful and respectful manner" on ln 379 or just take this sentence out.
14. ln 374, "The author assumes..." should it be the authors or if not, please clarify which author?
Author Response
Thank you for your insightful comments and suggestions. We appreciate your feedback and have made the necessary revisions to improve the manuscript
Comment 1: Please explain in the paper why each user only utilized the system once, given that you pointed out that reuse could change the perceived ease of use issues. (reference to use one month later)
Response 1: Our study aimed to assess the initial usability and ease of use of the social robot during a single session of interaction. This design allowed us to capture the participants' first impressions and unmediated reactions, which are crucial in identifying initial usability barriers that might not be as apparent in later sessions. Although reuse could indeed alter perceptions of ease of use, our focus on first-time interactions establishes a baseline understanding of how intuitive the system is for new users. Further studies could explore long-term use and how familiarity with the robot affects ease of use.
Comment 2: In the description of tasks, there are references to "stop the test" ... please describe in the paper if the "tasks" were explained as "tests" or "tasks" and whether the change in the terms led to confusion for the subjects. (this shows up in more than one place)
Response 2: Agree. On pages 7 and 8, as well as in Tables 1 and 2, we have replaced the terms 'Test' and 'Activity' with 'Task' where appropriate. Please note that in other sections, the term 'test' refers to the overall session of performing the tasks.
Comment 3: Please change the order of discussion of Table 2 and Table 1 or the order of the tables, so that the first discussion talks about the first table the user will find. (ie whatever the first table to be discussed should be called Table 1, etc)
Response 3: We prefer to change the order of the results and tables in the Result section. So that the coherence of the discussion will be maintained. We have repositioned them in the Result section (Page 11).
Comment 4: In the tables please put the legend for the table with the title of the table (which is at the top) rather than at the bottom of the table itself.
Response 4: We have revised our table legends and they are placed at the top of the tables. We have repositioned the captions for all figures to appear above them. (Pages 5 and 6).
Comment 5: Please clarify in the paper, whether "...the study employed ten people with MD and MCI..." (ln 111), involves 5 and 5 or 10 and 10.
Response 5: We have better defined the participant composition in the paper. “This mixed-methods study employed a total of ten people, 5 with MD and 5 with MCI...” (Page 6, line 8).
Comment 6: Please change micros to microphones (ln 229 and 314)
Response 6: We have changed this term to “microphones” (page 22,line 17 and Page 12, line 8)
Comment 7: Please check if Trial Phase (ln321) refers to a training phase; if it is the trial phase, please define in the paper what is intended.
Response 7: The term “trial” has been changed to “training”. (Page 17, line 28)
Comment 8: Please change the acronyms to be more consist with the words before them...eg "...not know the correct answer (DCR).." [use Correct Response or DCA?] etc; subtleties of the response method (DRM)... there are a number of these and aligning the words with the letters would be helpful. (or change the words to be consistent with the acronyms)
Response 8: "DCR stands for 'Difficulty with the Content of the Response,' and DRM stands for 'Difficulty in Recognizing the Response Method.' We intentionally kept the acronyms short for clarity, and we believe they are both appropriate.
Comment 9: please clarify in the paper who provided the "preliminary scores for SUS" and how this was done. (ln 189)
Response 9: We have added a short clarification to the section: “The participants completed the SUS at the end of the interaction session and the preliminary scores for the SUS were provided by the main researcher, who analyzed the data. “ (Page 9, Line 11)
Comment 10: In Figure 3, please provide bars to indicate the variability about the means
Response 10: The error bars for standard deviation have been added in Figure 3.
Comment 11: Please explain in the paper the basis for saying "...MINI's verbal communications rely on state of the art software..." (ln 356)
Response 11: We have made slight modifications to the sentence for better clarity and added a reference for that. “Given that MINI’s verbal communication relies on the available commercial software and solutions for speech recognition and production[14] …” (Page 19, Line 1)
Comment 12: Please supply the supplementary data referred to in ln 262. (the only supplementary data I found was the DEC 1986 questionaire; I'm assuming you are not referring to that.)
Response 12: We have added the missing supplementary material and changed the mistakes in naming them in the main text.
Comment 13: Please include in the paper support for the conclusion: "This comprehensive approach ensures social robots can effect effectively engage individuals with dementia in a meaningful and respectful manner" on ln 379 or just take this sentence out.
Response 13: We have removed the sentence and reordered the remaining sentences in the paragraph to improve coherence. “The authors assume that using the MINI robot effectively requires clear and detailed instructions and instructor assistance. By making suggested changes and listening to users, developers can improve the robot's usability and UX, ensuring the overall effectiveness. To overcome challenges, careful design and interaction methods, along with ongoing adjustments, are essential. This process can turn the MINI robot into a more user-centred system.” (Page 20)
Comment 14: ln 374, "The author assumes..." should it be the authors or if not, please clarify which author?
Response 14: The sentence has been corrected.
Reviewer 2 Report
Comments and Suggestions for Authors
This manuscript examines the MINI robot's UX and usability challenges, which is designed to assist seniors, offering cognitive and social engagement to elderly individuals affected by neurodegenerative disorders, particularly those dealing with cognitive decline.
The testing and analysis process of the manuscript is relatively complete. There is one confusion is that only the chosen tasks and interaction methods are introduced and adopted in this paper. However, the reasons for choosing these methods and tasks are not given.
It is recommended to give an appropriate and strong explanation of why these methods were chosen.
1. About the methods. The authors did not explain why the tasks and interaction methods are chosen in line 140. It is recommended to give an appropriate and strong explanation of why these methods were chosen.
2. About the extensibility of the research method. Are there universal research methods for this class of interactive robots that can suggest improvements for the “User Experience and Usability” of these interactive robots? For example, in order to give the suggested modifications of these robots, Is there a universal way to interact with these robots?
3.About the literature review. In section 2.3, the System Usability Scale (SUS) tool is used to evaluate the perceived usability of MINI robot, which is widely used in diverse products. However, there is a lack of specific cases of its wide application in the introduction.
Author Response
Evaluating the User Experience and Usability of the MINI Robot for Elderly Adults with Mild Dementia and Mild Cognitive Impairment: Insights and Recommendations.
Response to Reviewer 2 Comments
Thank you for your insightful comments and suggestions. We appreciate your feedback and have made the necessary revisions to improve the manuscript.
Comment 1: About the methods. The authors did not explain why the tasks and interaction methods are chosen in line 140. It is recommended to give an appropriate and strong explanation of why these methods were chosen.
Response 1: We selected these tasks based on the fundamental interactions a user would need to effectively interact with the robot, such as initiating, pausing, and responding to it. The tasks were determined through discussions between the developers and researchers, following several hands-on sessions with the robot to identify key functionalities essential for user interaction.
Comment 2: About the extensibility of the research method. Are there universal research methods for this class of interactive robots that can suggest improvements for the “User Experience and Usability” of these interactive robots? For example, in order to give the suggested modifications of these robots, is there a universal way to interact with these robots?
Response 2: There is no strict universal method of interaction for this class of interactive robots, as their functions and features can vary significantly based on their target population and intended purpose. However, some robots are similar in certain functions and we believe our recommendations for improvements could be useful for them, as well as the Mini’s developers itself.
Comment 3: About the literature review. In section 2.3, the System Usability Scale (SUS) tool is used to evaluate the perceived usability of the MINI robot, which is widely used in diverse products. However, there is a lack of specific cases of its wide application in the introduction.
Response 3: We included a statement highlighting the SUS as the most common used questionnaire in the field of social robots: 'According to a recent scoping review [18], the SUS is the most commonly used questionnaire for evaluating the usability of social robots' (Page 9, line 7)."
Reviewer 3 Report
Comments and Suggestions for Authors
The idea to evaluate the usability of the MINI robot with mild neorodegenerative disorder adults is interesting. However, the sample size is only 10 participants and may not be representative of the real cases. The procedure is clear and well conducted. SUS provides a good quantitative measure. Hardware of the MINI robot sould be more revealed/explained.
Author Response
Thank you for your thoughtful feedback. While we acknowledge that the sample size of 10 participants may seem limited, it is appropriate for initial testing according to SUS guidelines and existing literature. We believe we have provided sufficient explanation of the MINI robot's hardware for the context of this study, but we appreciate your suggestion and will consider enhancing this aspect in future studies.